# Investigation of Biodegradation and Biocompatibility of Chitosan–Bacterial Cellulose Composite Scaffold for Bone Tissue Engineering Applications

**DOI:** 10.3390/cells14100723

**Published:** 2025-05-15

**Authors:** Somchai Yodsanga, Supattra Poeaim, Soranun Chantarangsu, Somporn Swasdison

**Affiliations:** 1Department of Biology, School of Science, King Mongkut’s Institute of Technology Ladkrabang (KMITL), Ladkrabang, Bangkok 10520, Thailand; 2Department of Oral Pathology, Faculty of Dentistry, Chulalongkorn University, Bangkok 10330, Thailand; soranun.c@chula.ac.th; 3Department of Oral Medicine, College of Dental Medicine, Rangsit University, 52/345 Phahonyothin Rd., Mueang Pathum Thani District, Pathum Thani 12000, Thailand; ssomporn@chula.ac.th

**Keywords:** CS–BC composite scaffold, solvent casting/particulate leaching method, osteoblastic differentiation, in vitro studies

## Abstract

Developing scaffolds with a three-dimensional porous structure and adequate mechanical properties remains a key challenge in tissue engineering of bone. These scaffolds must be biocompatible and biodegradable to effectively support osteoblastic cell attachment, metabolic activity, and differentiation. This study successfully fabricated a chitosan–bacterial cellulose (CS–BC) composite scaffold using the solvent casting/particle leaching (SCPL) technique, with NaOH/urea solution and sodium chloride crystals as the porogen. The scaffold exhibited a well-distributed porous network with pore sizes ranging from 300 to 500 µm. Biodegradation tests in PBS containing lysozyme revealed a continuous degradation process, while in vitro studies with MC3T3-E1 cells (pre-osteoblastic mouse cell line) demonstrated excellent cell attachment, as observed through SEM imaging. The scaffold also promoted increased metabolic activity (OD values) in the MTT assay, and enhanced alkaline phosphatase (ALP) activity and upregulated expression of osteogenic-related genes. These findings suggest that the CS–BC composite scaffold, fabricated using the SCPL method, holds great potential as a candidate for bone tissue engineering applications.

## 1. Introduction

Bone tissue engineering provides an effective approach for restoring bone lost or damaged due to injury, disease, or surgical procedures. A key component of this approach is the design of a porous scaffold that provides a three-dimensional environment, facilitating nutrient and oxygen delivery while promoting cell attachment, growth, and differentiation for new bone tissue formation. The ideal scaffold must be biocompatible, supporting normal cellular processes without causing toxicity. It should also be biodegradable, maintain adequate mechanical strength, and feature pores larger than 300 µm to support cell penetration and bone tissue growth [1,2]. In recent years, natural polymers like collagen, gelatin, alginate, and chitosan have gained popularity for scaffold creation in bone tissue engineering. These polymers are favored for their excellent biocompatibility, biodegradability, and non-toxic nature, and they offer structural characteristics similar to the native extracellular matrix, enabling enhanced cell interaction, attachment, and growth [3].

Chitosan (CS) is widely regarded as a key natural polymer for scaffold fabrication in bone tissue engineering [4]. It consists of linear chains of glucosamine and N-acetylglucosamine units connected by β (1–4) glycosidic bonds. CS is obtained by deacetylating chitin, a substance found in the shells of crustaceans like shrimp and crabs [5]. CS closely resembles glycosaminoglycans, key bone and cartilage extracellular matrix components, where they interact with collagen fibers and support cell–cell adhesion [6,7]. CS scaffolds have demonstrated the ability to support osteoblastic cell attachment and proliferation, exhibiting strong osteoconductivity that promotes bone growth both in vitro and in vivo studies [8,9]. CS scaffolds have limited mechanical strength despite their benefits, restricting their use in load-bearing applications. Combining CS with other polymers to create composite scaffolds has become a promising solution to address this limitation. This approach has generated significant attention for its potential in advancing new applications [10,11].

Bacterial cellulose (BC) is a white gelatinous substance produced by specific bacteria such as *Acetobacter xylinum* [12]. The chemical structure of BC is similar to CS, with the key difference being that BC contains a hydroxyl group at the C-2 position on the glucose molecule rather than the amino group found in CS [13]. BC offers several valuable properties, including excellent biocompatibility, a highly porous three-dimensional nanofibrillar structure, superior mechanical strength in dry and wet conditions, and an impressive ability to retain water. These attributes make BC an ideal material for biomedical applications, such as drug delivery, artificial skin, blood vessels, wound care, and tissue engineering scaffolds [14].

The combination of chitosan (CS) and bacterial cellulose (BC) has gained considerable attention because their structural similarities lead to composite materials that harness the biological benefits of CS and the enhanced mechanical strength of BC [15]. Various studies have explored the production of CS–BC composites, such as adding CS to the culture medium during BC biosynthesis or immersing BC in CS solutions. These approaches have demonstrated that CS can integrate with BC microfibrils, forming a denser network structure within the composite material [16,17]. Additionally, CS molecules can occupy the voids in the BC network, fostering strong interactions between the two components [18]. Although CS–BC composite materials show improved mechanical properties, enhanced cell attachment, and better cell growth compared to individual BC or CS, the pore size in the composite materials tends to decrease relative to that of natural BC. This is attributed to the incorporation of CS into the BC network [19,20]. The pore size of natural BC pellicles is typically under 100 nm, which can hinder cell infiltration and migration [21].

Various methods for creating polymer scaffolds have been developed, such as solvent casting/particulate leaching (SCPL), freeze-drying, gas foaming, electrospinning, and phase separation [22]. SCPL is the most commonly used method due to its straightforward process. The process involves dissolving a polymer in a solvent, combining it with a porogen such as sodium chloride (NaCl) crystals, molding the mixture, and then leaching the porogen and solvent to create a porous scaffold [23]. One of the key benefits of SCPL is that it allows precise control over the porosity and pore size of the scaffold by adjusting the quantity and particle size of the NaCl crystals incorporated into the mixture [24]. Our previous work successfully fabricated a CS–BC composite scaffold with a three-dimensional porous structure using the SCPL method. We dissolved CS and BC in a sodium hydroxide (NaOH)/urea solution and added NaCl crystals (450–500 μm) to achieve the desired porosity and pore size. The resulting CS–BC composite scaffold had a porosity of about 92% and pore sizes averaging 300–500 μm, along with strong mechanical properties and high-water absorption [25]. However, its biocompatibility and biodegradability still need to be thoroughly investigated to evaluate how the incorporation of CS with BC affects the scaffold’s performance, which is essential for determining its suitability for bone tissue engineering. Therefore, this study aims to evaluate the in vitro biodegradation and osteogenic differentiation of MC3T3-E1 cells on the CS–BC composite scaffold prepared via the SCPL method by analyzing cell attachment, the metabolic activity of cells, and osteogenic gene expression using real-time quantitative polymerase chain reaction (RT-qPCR).

## 2. Materials and Methods

### 2.1. Fabrication of CS–BC Composite Scaffold

In our previous report, we successfully fabricated a porous CS–BC composite scaffold with a 1:1 weight ratio using the SCPL method. The process began by dissolving 0.1 g of chitosan (CS) powder (extracted from shrimp shells, ≥75% deacetylated, C3646-25G, Sigma-Aldrich, St. Louis, MO, USA) in 10 mL of a NaOH/urea/water solution pre-cooled to −12 °C (weight ratio 7:12:81, KA482 and KA817, KEMAUS, Cherrybrook, NSW, Australia) to prepare a 1 wt% CS solution. This solution underwent six freeze–thaw cycles to improve its consistency. Next, 0.1 g of bacterial cellulose (BC) powder was added and gently mixed at −12 °C to achieve a uniform solution. Following this, 8 g of sodium chloride (NaCl) crystals (KA465, KEMAUS, Cherrybrook, NSW, Australia), with particle sizes between 450 and 500 μm, were added and evenly distributed throughout the CS–BC mixture. The prepared solution was transferred into a glass tube and left to solidify. It was then immersed in distilled water multiple times until the pH reached 7. The sample underwent overnight freezing at −20 °C, followed by a 24 h freeze-drying process at −50 °C using a freeze-dryer (Millrock Technology Inc., New York, NY, USA).

### 2.2. Scanning Electron Microscopy

Scanning electron microscopy (SEM, FEI Quanta 250, Eindhoven, The Netherlands) was used to observe the porous structural scaffolds. Cross-sectional scaffolds were mounted on the adhesive aluminum stubs and covered with a thin gold layer using a modular coater system (Quorum Model Q150R, Laughton, UK). The images were captured using an acceleration voltage of 15 to 20 kV to evaluate detailed surface and structural characteristics.

### 2.3. In Vitro Biodegradable Study

The scaffold’s biodegradability was assessed through an in vitro biodegradation study. The samples were immersed in 5 mL of PBS-lysozyme solution (pH 7.4) containing 10,000 U/mL lysozyme from Sigma-Aldrich (L1667, St. Louis, MO, USA), kept under 37 °C in an incubator, and the fresh solution was replaced every 3 days [26]. The scaffold’s initial dry weight was denoted as *Wi*. After being incubated for 7, 14, and 21 days, the scaffolds were removed from the solution, rinsed gently with distilled water, and dried in an oven at 35 °C for 5 days. The dried scaffolds were again weighted and denoted as *Wt*. The scaffolds submerged in PBS solution without lysozyme were also designed as the control group. The percentage of weight loss (*W_L_*%) due to the scaffold’s biodegradability was determined using the following equation:*W_L_*% = [(*Wi* − *Wt*)/*Wi*] × 100(1)

### 2.4. In Vitro Biocompatible Study

#### 2.4.1. Cell Culturing and Seeding Procedure

The pre-osteoblastic mouse cell line, MC3T3-E1, was cultured in a growth medium supplemented with alpha-MEM (SH30265.02, Cytiva, HyClone Laboratories, Logan, UT, USA), 10% fetal bovine serum (FBS, 1600044, Gibco, Waltham, MA, USA), and 1% antibiotic mix (penicillin/streptomycin, 15240062, Gibco, Waltham, MA, USA), and maintained in a 5% CO_2_ incubator at 37 °C. Before testing, the circular discs of scaffolds (15 × 1 mm) were sterilized using 70% ethanol and UV light for 10- and 20-min. Sterilized scaffolds were then placed into the 24-well tissue culture plates, pre-incubated in growth medium at 37 °C overnight, and seeded with 3.5 × 10^4^ cells per well directly onto each scaffold in triplicate. Control wells without scaffolds were also included for comparison. The fresh growth medium was replaced every two days.

#### 2.4.2. Cell Attachment

SEM examined the MC3T3-E1 cells attached to CS–BC scaffolds following incubation for 1, 3, and 7 days, with preparation steps including PBS washing, fixation, dehydration, drying, and gold sputter-coating.

#### 2.4.3. Metabolic Activity of Cells

The MTT assay (M6494, ThermoFisher Scientific, Waltham, MA, USA) was performed to evaluate the metabolic activity of cells on CS–BC scaffolds at determined time points (on days 1, 3, and 7). After washing in PBS solution, each sample well was treated with 500 µL of MTT solution (0.5 mg/mL in phenol red-free DMEM) and incubated in the dark at 37 °C for 4 h. Afterward, the MTT solution was replaced with dimethylsulfoxide (DMSO) to dissolve the formazan crystals, and the optical density (OD) values were measured in triplicate at 570 nm.

### 2.5. Cell Differentiation

Following 24 h of seeding, the growth medium was replaced with an osteogenic medium to promote osteoblastic differentiation. The osteogenic medium comprised of the growth medium, 50 µg/mL L-ascorbic acid (A92902, Sigma-Aldrich, St. Louis, MO, USA), 10 nM dexamethasone (D4902, Sigma-Aldrich, St. Louis, MO, USA), and 10 nM β-glycerophosphate (G9422, Sigma-Aldrich, St. Louis, MO, USA). The cultures were set at 37 °C in a 5% CO_2_ incubator, maintained for 7, 14, and 21 days, and the fresh medium was renewed every 2 days.

#### 2.5.1. Alkaline Phosphatase (ALP) Enzyme Activity Assay

ALP activity, which indicates osteoblastic differentiation, was measured using the BCIP/NBT substrate system (203790, Sigma-Aldrich, Darmstadt, Germany) following the protocol of Wang et al. [27], with OD readings taken at 405 nm.

#### 2.5.2. Gene Expression Analysis by RT-qPCR

Cellular RNA was isolated from MC3T3-E1 cells in triplicate composite scaffolds using TRIzol reagent (Invitrogen, Carlsbad, CA, USA). cDNA synthesis was performed using a reverse-transcription kit (iScript Reverse Transcription Supermix, Bio-Rad, Hercules, CA, USA), following the supplier’s protocol. A thermal cycler (T100, Bio-Rad Laboratories Inc., Hercules, CA, USA) performed the reverse-transcription condition at 25 °C for 5 min, 46 °C for 20 min, and 95 °C for 1 min. Osteoblastic differentiation was evaluated by RT-qPCR using the FastStart Essential DNA Green Master Mix kit (Roche Diagnostics, Mannheim, Germany). The reactions were operated with a LightCycler 480 II system (Roche Diagnostics, Mannheim, Germany) and the following settings: 45 cycles of 95 °C for 30 s, 58 °C for 30 s, and 72 °C for 30 s. Negative controls without cDNA were included to ensure specificity. Specific primers for *osteocalcin (OCN)*, *alkaline phosphatase (ALP)*, *collagen type I (COL-1)*, and *bone sialoprotein (BSP)* were used to check osteogenic-related genes, which were normalized against the expression level of *glyceraldehyde-3-phosphate dehydrogenase (GAPDH*, as internal control) with results expressed as fold changes compared to controls [28]; all primers are listed in Table 1.

### 2.6. Statistical Analysis

Each experiment was conducted in triplicate, and the data are expressed as mean values with corresponding standard deviations. The Shapiro–Wilk test assessed data normality, while Levene’s test was applied to evaluate variance homogeneity. For statistical comparisons, an independent t-test was employed for two-group analyses, while one-way ANOVA with Bonferroni post hoc tests was used for multi-group comparisons. IBM SPSS Statistics version 29.0 (IBM Corp., Armonk, NY, USA) was used to conduct the statistical analyses, and all graphical data visualizations were created with GraphPad Prism 8 (GraphPad Software, San Diego, CA, USA). A *p*-value of less than 0.05 was considered statistically significant.

## 3. Results

### 3.1. SEM Analysis of CS–BC Composite Scaffold Morphology

The CS–BC composite scaffold fabricated using the SCPL method is depicted in Figure 1a, showcasing a three-dimensional porous sponge structure after the freeze-drying process. The SEM image in Figure 1b illustrates the cross-section of the freeze-dried CS–BC composite sponge, revealing a porous network with uniformly distributed pores and rough pore walls. The pore sizes ranging from approximately 300 to 500 µm were revealed. These structural characteristics suggest that the CS–BC composite scaffold possesses favorable morphology and pore size.

### 3.2. The Biodegradation of CS–BC Scaffold

The biodegradation behavior of the developed CS–BC composite scaffolds was evaluated through an in vitro study. As shown in Figure 2, the results reveal a gradual increase in weight loss across all groups throughout the degradation period. Precisely, the weight loss of the CS–BC composite scaffolds was measured at 10% (±1.15), 18% (±0.00), and 24% (±0.58) on days 7, 14, and 21, respectively. The control group showed weight loss of 2% (±0.00), 5% (±0.00), and 8% (±1.15) at the respective time points, demonstrating that the CS–BC composite scaffolds experienced significantly more significant weight loss. The results confirm the biodegradability of CS–BC scaffolds in the presence of lysozyme, with their degradation rate dependent on the incubation duration.

### 3.3. MC3T3-E1 Cell Attachment and Metabolic Activity on the CS–BC Composite Scaffold

The evaluation of MC3T3-E1 cell attachment and the metabolic activity of cells on the CS–BC scaffold are depicted in Figure 3a–c. On day 1 post-seeding, SEM images reveal individual cells with a spherical morphology beginning to adhere to the pore walls of the composite scaffold (Figure 3a). The cells appeared evenly distributed across the surface, indicating efficient seeding. The initial spherical shape suggests that the cells were in the early stages of attachment as they began establishing interactions with the scaffold material. By day 3, the cells exhibited an irregular, rounded morphology with visible pseudopodia extending to anchor themselves to the scaffold surface (Figure 3b). These projections indicate active cell-scaffold interaction, crucial for promoting cellular adhesion and subsequent proliferation. The cells appeared to bridge across the pores, demonstrating the scaffold’s ability to support cellular spreading and integration. By day 7, the cells displayed a flattened shape and remained securely attached to the pore walls of the scaffold (Figure 3c). This morphological change suggests that the cells had adapted to the scaffold’s surface and achieved stable adhesion. Additionally, the cells showed signs of forming a continuous layer, potentially indicating the onset of extracellular matrix production.

The metabolic activity of cells was assessed quantitatively using the MTT assay, with results in Figure 4. The optical density (OD) values of the cells cultured on the composite scaffold increased progressively throughout the experiment, indicating enhanced metabolic activity over time. On day 1, the OD value for MC3T3-E1 cells cultured on the CS–BC composite scaffold was significantly lower than the control group, suggesting that the initial adhesion phase required an adjustment period as the cells adapted to the scaffold’s surface. However, by day 3, the OD value for the composite scaffold group showed a marked increase, surpassing the control, highlighting the scaffold’s supportive environment for cellular proliferation. By day 7, the OD values of both groups were comparable, with no statistically significant difference (ns), indicating consistent cell viability across both conditions. The results of the MTT assay (Figure 4) underscore the CS–BC composite scaffold’s ability to sustain cellular attachment and support metabolic activity over time. The observed increase in OD values from day 1 to day 7 reflects the scaffold’s biocompatibility and capacity to support the metabolic activity of MC3T3-E1 cells. These findings further confirm that the scaffold provides a favorable microenvironment for cellular growth. This makes it a potential candidate for bone tissue engineering applications.

### 3.4. MC3C3-E1 Cell Differentiation on the CS–BC Composite Scaffold

Early markers of osteoblastic differentiation, such as ALP enzyme activity, reflect the progression of the osteogenic process. Figure 5 presents the ALP activity of MC3T3-E1 cells cultured on the CS–BC composite scaffold under osteogenic induction conditions over 21 days. The optical density (OD) values show a time-dependent increase in ALP activity, highlighting the scaffold’s supportive role in cell differentiation. On day 7, the ALP activity was relatively low, suggesting that the cells were in the early stages of differentiation. By day 14, a notable increase in ALP activity was observed, indicating that the cells were advancing toward osteoblastic maturation. On day 21, ALP activity peaked, approximately doubling compared to the levels recorded on day 7 and showing a further increase compared to day 14. These findings emphasize the CS–BC composite scaffold’s potential to promote and sustain osteoblastic differentiation effectively over time.

The gene expression analysis reveals a gradual increase in key osteogenic markers over 21 days. The mRNA levels of *OCN*, *ALP*, *COL-1*, and *BSP* progressively elevated in the scaffold group and were consistently higher than in the control group, as shown in Figure 6a–d. *OCN* expression remained similar to the control on day 7, then significantly increased by day 14, peaking at day 21 (Figure 6a). *ALP* showed no significant difference on day 7 but rose significantly by day 14 and continued to increase through day 21 (Figure 6b). *COL-1* exhibited a moderate yet significant increase on day 7, further intensifying on day 14 and reaching a marked elevation by day 21 (Figure 6c). Likewise, *BSP* expression was comparable to the control on day 7 but significantly increased on day 14 and peaked on day 21 (Figure 6d). Overall, these results suggest that the CS–BC scaffold supports osteogenic differentiation effectively.

## 4. Discussion

A suitable scaffold for tissue engineering of bone should offer a well-defined three-dimensional porous structure and sufficient mechanical stability. In addition, it must exhibit biodegradability and biocompatibility, ensuring non-toxicity while supporting cell adhesion, cell division, cell metabolic activity, and cell differentiation toward the formation of new bone tissue [29].

To the best of our knowledge, this is the first report that combines a NaOH/urea–based SCPL technique to fabricate CS–BC composite scaffolds with highly interconnected macropores suitable for bone tissue engineering. While individual use of CS or BC has been previously explored, the synergistic integration using this fabrication strategy offers a novel approach that improves porosity, mechanical strength, and biocompatibility.

The SCPL method has proved successful in fabricating bone tissue engineering scaffolds with various polymers. This method allows for the control of structural characteristics such as porosity and pore sizes by incorporating amounts and a known diameter of particles into polymer solutions [30]. Compared to other scaffold fabrication methods, it is inexpensive and relatively simple to create a high porosity scaffold with pore diameters up to 500 um [31]. In this study, the main goal of implementing this method is to combine CS with BC via a low-temperature NaOH/urea solution and optimize the porous structure with pore size in the composite scaffold of CS–BC by NaCl crystals. The results of SEM analysis reveal a porous structure, pore size (300 to 500 um in range), a connected pore network, and consistent pore distribution with surface roughness in the CS–BC composite scaffold, which were determined by the size of the NaCl particulate template. Khan et al. reported that the high porous structural characteristics with the pore sizes in scaffolds were based on the amount and size of NaCl crystals added in the polymer solution [32,33]. However, the high porosity and the large pore sizes may affect the low mechanical properties of the scaffolds [34]. The compressive strength of spongy bone is in the range of 0.2–4 MPa [35]. Our previous studies have shown that the compressive strength of the CS–BC composite scaffold was approximately 1.3 MPa in the dry state and 0.62 MPa in the wet state [25]. These compressive strengths of scaffolds are good enough to be used in non-load-bearing sites of bone tissue [36]. This may be due to the homogenous distribution of CS and BC particles in NaOH/urea solution and abundant hydroxy groups in the chemical structure of BC, which provides an excellent incorporation of CS molecules. This led to the mechanical properties of the CS–BC composite scaffold [37,38].

The CS–BC composites are predominantly fabricated through two strategies: (i) incorporating chitosan during in situ bacterial cellulose biosynthesis and (ii) post-synthetic immersion of BC membranes in acetic acid solutions containing chitosan. In these hybrid matrices, chitosan molecules infiltrate the nanofibrillar cellulose network and establish robust intermolecular hydrogen bonding with cellulose chains, reinforcing the mechanical and structural integrity of the scaffold. However, a critical limitation of conventional CS–BC composites lies in their restricted porosity (typically <1 μm), as chitosan preferentially occupies the interfibrillar voids, thereby impeding cellular infiltration and limiting osteogenic tissue ingrowth [39,40,41].

We developed a highly porous CS–BC scaffold using the SCPL method to overcome this limitation. NaOH/urea solution served as the dissolution medium for both components and salt crystals functioned as porogen. The NaOH/urea system facilitates homogeneous dispersion and integration of CS and BC, enabling successful scaffold fabrication. Importantly, this solvent system is cost-effective, non-toxic, and environmentally benign, aligning with the principles of green chemistry [25].

Given their structural similarity as polysaccharides, both CS and BC have been shown to dissolve in NaOH/urea aqueous systems, particularly at low temperatures. A solvent system comprising NaOH/urea/H_2_O in a 7:12:81 weight ratio has proven effective for individually dissolving each polymer. Pandey et al. demonstrated that 2 g of BC powder could be fully solubilized within minutes in a −12 °C pre-cooled solution [38], while Chang et al. achieved complete dissolution of 1 g of CS in the same solvent at −20 °C using a freeze–thaw cycle, yielding a transparent chitosan solution [42]. Mechanistically, NaOH disrupts the intra- and inter-molecular hydrogen bonds within the polymer chains. In contrast, under subzero temperatures, urea inhibits re-aggregation by forming hydrogen bonds with the polymer backbones, stabilizing the dissolved state. Although this NaOH/urea binary solvent is considered environmentally benign and effective for the dissolution of CS and BC, achieving a homogeneous and stable integration of both polymers in a single solution remains a significant challenge, largely due to differences in molecular weight, solubility kinetics, and potential phase separation during scaffold fabrication [43,44,45].

Our previous work demonstrated that a 1:1 (*w*/*v*) ratio of CS to BC could be processed into a homogeneous blend using a low-temperature NaOH/urea aqueous system. The CS solution was first obtained via six freeze-thaw cycles in a pre-cooled NaOH/urea solution. BC was introduced and uniformly dispersed into the CS solution at −12 °C until a translucent mixture was achieved. Upon casting in the presence of salt crystals, the solution underwent thermally induced gelation at room temperature. Subsequent immersion in distilled water effectively removed residual NaOH/urea and salt, yielding a purified CS–BC hydrogel scaffold. Fourier-transform infrared (FTIR) spectroscopy confirmed the presence of characteristic functional groups from both CS and BC, supporting the successful incorporation of both polymers within the composite matrix [25]. These observations are consistent with previous findings from CS–BC membranes prepared either via in situ biosynthesis in chitosan-supplemented media or by post-synthetic immersion in chitosan solution, both of which showed evidence of intermolecular hydrogen bonding and nanofibrillar integration [46,47]. These results support the idea that the CS–BC scaffold fabricated herein likely derives its structural integrity from a dense hydrogen bonding network between the constituent polysaccharide chains.

Biodegradation is a critical property of scaffolds for bone tissue engineering, requiring them to gradually degrade to create space for new bone growth and matrix deposition following cell proliferation [48]. Although CS and BC have been studied for their beneficial properties for bone regeneration, it is well known that CS-based scaffold alone is quickly degraded. In contrast, natural BC-based scaffold is not degraded in the human body due to the absence of cellulose-degrading enzymes. This is one of their limitations for applications in bone tissue engineering [49,50]. This work assessed the in vitro biodegradation of CS–BC composite scaffolds using a PBS solution containing lysozyme, an enzyme naturally present in various human and animal tissues and fluids [51]. The results show a consistent reduction in the scaffolds’ weight over time, indicating their biodegradability in the presence of lysozyme. This degradation behavior is likely due to the β (1→4) glycosidic bonds between glucosamine and N-acetylglucosamine in chitosan, which lysozyme can cleave [52]. Additionally, bacterial cellulose may contribute to the degradation process due to its structural instability, reduced crystallinity, and the morphological characteristics of the scaffold [53]. The findings suggest that the CS–BC composite scaffold has a prolonged degradation rate, making it a strong contender for bone tissue engineering applications.

The biocompatibility of scaffolds is essential in bone tissue engineering, as it directly influences cellular adherence, cell division, proliferation, and differentiation [54]. In this study, the CS–BC composite scaffold demonstrated excellent biocompatibility with MC3T3-E1 cells, supporting their attachment and proliferation without exhibiting cytotoxic effects. Jianqing et al. reported that adding CS to a silk fibroin–gelatin composite scaffold improved the adhesion and proliferation rate of MC3T3-E1 [55], while Min et al. reported that CS enhanced osteoinductive properties in hyaluronic–collagen composite scaffold [56]. The scaffold’s natural composition, derived from CS and BC, and its extracellular matrix-like structure significantly enhanced osteoblastic attachment and proliferation [57,58]. CS has a similar structure to glycosaminoglycans and can provide a suitable microenvironment for the attachment and proliferation of osteoblastic cells. At the same time, BC possesses a structural resemblance to collagenous fibers in bone tissue, which provide strength and structural support for cell attachment and interaction [59,60]. These properties underline the suitability of CS–BC scaffolds for bone tissue applications. Furthermore, the scaffold’s ability to promote early osteoblastic differentiation was demonstrated by increased ALP activity and the showing of osteogenesis-related genes [61]. In addition, Yu et al. found that CS–hydroxyapatite could stimulate the expression of ALP activity and osteogenic genes of bone-forming cells better than a hydroxyapatite-based scaffold alone [9]. These findings suggest that CS–BC composite scaffolds hold promise as a viable substrate for promoting bone regeneration in tissue engineering applications.

The initiation of osteoblastic differentiation in MC3T3-E1 cells is marked by increased ALP activity, a critical process for promoting bone mineralization [62]. In this study, the CS–BC composite scaffold demonstrated the ability to enhance ALP activity over time, confirming its role in supporting the differentiation of MC3T3-E1 cells. The scaffold also facilitated the upregulation of key osteogenic-related genes, including *OCN*, *ALP*, *COL-1*, and *BSP*, which are essential for various stages of osteoblastic differentiation [63,64]. *OCN*, pivotal in bone matrix formation, showed elevated expression, indicating the transition of cells to the mineralization phase [65]. *ALP* plays a key role in extracellular matrix mineralization, with its expression rising during differentiation, serving as a marker for osteoblast differentiation [66]. *COL-1*, the primary component of the bone matrix, is produced during the differentiation of osteoblasts [67]. Moreover, *BSP* is a marker for late differentiation of osteoblasts and a gene related to the mineralization phase of bone formation and support cell attachment [68]. The relative expression level of *OCN*, *ALP*, *COL-1*, and *BSP* had been observed in the in vitro studies in various composite scaffolds such as CS/hydroxyapatite/collagen, hydroxyapatite/CS/gelatin, BC/collagen, BC/gelatin/hydroxyapatite, and polyglycolide/polycaprolactone. These gene expressions were found to be low on the initial time, but they were significantly upregulated over time and demonstrated notable differences when compared to the control group [69,70,71,72,73]. Interestingly, it was also observed that adding CS- or CS and BC-based scaffolds incorporated with other ceramics or polymers excellently promoted the differentiation of osteoblastic cells [74,75]. Our findings reveal that the expression levels of osteogenic-related genes, *OCN*, *ALP*, *COL-1*, and *BSP*, in MC3T3-E1 cells cultured on the CS–BC composite scaffold consistently increased over time and were considerably more remarkable than the control group. These results highlight the ability of the CS–BC composite scaffold to enhance the osteoblastic differentiation of MC3T3-E1 cells.

Although this study did not directly investigate the underlying cellular mechanisms, accumulating evidence indicates that CS facilitates the adhesion and proliferation of osteoblasts and mesenchymal stem cells through electrostatic interactions between its protonated amino groups and negatively charged cell membranes [76]. These amino functionalities also engage with glycosaminoglycans and proteoglycans in the extracellular matrix, promoting the release of cytokines and osteogenic growth factors essential for bone regeneration [77]. Conversely, the BC provides a highly porous nanofibrillar architecture resembling native collagen, thereby supporting cell adhesion, migration, and nutrient exchange [78]. Combining CS and BC into composite scaffolds has recently shown promise in bone tissue engineering applications. These scaffolds support osteoblast adhesion and proliferation and enhance osteogenic differentiation, as evidenced by upregulated ALP activity and increased expression of osteogenic markers such as *ALP*, *OCN*, *COL-1*, and *BSP* [72,79,80].

The findings above underscore the favorable biodegradability and cytocompatibility of the CS–BC composite scaffold in vitro, supporting its potential utility in bone tissue engineering. A key limitation of this study is the absence of in vivo validation. Nevertheless, previous studies have demonstrated that scaffolds composed of CS and BC exhibit distinct degradation profiles and regenerative capabilities in animal models. CS-based scaffolds undergo enzymatic degradation primarily via lysozyme, an endogenous enzyme abundant in mammalian lymphoid tissues [81]. The resulting degradation byproducts, including D-glucosamine and N-acetylglucosamine, are biocompatible and readily metabolized or excreted. The degradation kinetics of CS are influenced by its molecular weight and degree of deacetylation, with lower molecular weight and deacetylation correlating with faster degradation rates [82,83]. In contrast, BC exhibits high crystallinity and lacks susceptibility to enzymatic hydrolysis in vivo, owing to the absence of cellulase enzymes in mammals [84]. Consequently, BC degrades slowly, if at all. In a rat calvarial defect model, BC scaffolds remained largely intact 12 weeks post-implantation, which partially impeded new bone formation [72]. These findings suggest that while CS contributes to biodegradability, BC’s recalcitrance to degradation remains challenging for long-term in vivo applications. Future scaffold designs may require chemical modification or enzymatically susceptible analogs to improve BC degradability without compromising structural integrity.

Successful bone tissue regeneration hinges on the presence of osteogenic progenitor cells and growth factors and the design of porous scaffolds capable of supporting vascularization. Vascular networks are essential for transporting oxygen and nutrients, removing metabolic byproducts, and directly modulating osteogenic processes via endothelial cell signaling [85]. Despite the increasing interest in CS–BC composite scaffolds, their angiogenic potential remains underexplored. Recent in vivo studies, however, suggest that CS–BC composites may promote angiogenesis [18]. Lin et al. observed the formation of dense vascular networks surrounding and penetrating CS–BC implants, in contrast to minimal vascularization observed with native BC membranes [15]. Similarly, Li et al. reported the accumulation of endothelial cells near the interface of CS–BC scaffolds, implying a role in guiding neovascularization [86]. This pro-angiogenic effect may be attributed to the cationic nature of CS, which is rich in protonated amino groups that can electrostatically interact with anionic components of angiogenic signaling pathways, including glycosaminoglycans and vascular endothelial growth factor (VEGF) [87]. These findings underscore the potential of CS–BC composites as structural scaffolds and bio-interactive platforms capable of modulating vascular responses. However, mechanistic studies dissecting the molecular basis of CS-induced angiogenesis remain limited and merit further investigation.

One of the central challenges in bone tissue engineering is the design of composite scaffolds that exhibit a balanced combination of mechanical integrity, biodegradability, biocompatibility, and appropriate porosity [88]. In the present study, the CS–BC composite scaffold demonstrated susceptibility to enzymatic degradation by lysozyme and effectively supported the adhesion, proliferation, and osteogenic differentiation of MC3T3-E1 preosteoblasts. These results are consistent with previous reports indicating the bioactivity of CS–BC matrices. However, prior studies have also highlighted the non-biodegradability of BC owing to its highly crystalline structure, which limits its resorption in vivo [89,90]. Although in vivo evaluations were beyond the scope of this study, the observed in vitro performance positions CS–BC scaffolds as promising candidates for bone regeneration, comparable to established composites such as collagen–hydroxyapatite (COL–HA). Incorporating collagen into HA scaffolds enhances their mechanical performance while conferring osteoinductive and osteoconductive properties due to the inherent bioactivity of both components [91]. Similarly, CS–BC composites have been shown to promote osteogenic differentiation, suggesting that they, too, possess osteoconductive and osteoinductive characteristics [92,93]. Notably, BC enhances the mechanical robustness of CS-based scaffolds, paralleling synthetic polymer-based systems such as polylactic acid (PLA), polyglycolic acid (PGA), and polycaprolactone (PCL), but with the added advantage of natural origin and ECM-like structure [94]. Importantly, CS and BC share structural similarities with native extracellular matrix components, facilitating favorable cell-material interactions. Comparative studies have reported that CS–BC composites outperform several natural polymer-based scaffolds—including collagen, silk fibroin, gelatin, and alginate—in promoting osteoblast activity and bone tissue formation [95,96]. These findings collectively underscore the potential of CS–BC scaffolds as bioinspired platforms for bone tissue engineering.

Despite the promising in vitro outcomes, several limitations of the present study should be noted. First, the CS–BC composite scaffold evaluation was limited to in vitro assays and did not include direct comparisons with clinically approved or commercially available scaffolds. Comparative studies with standard bone graft substitutes are necessary to contextualize the CS–BC composites’ biodegradability, biocompatibility, and mechanical properties. Second, while osteoconductivity was demonstrated under osteogenic culture conditions, osteoinductivity, an essential attribute for bone scaffolds, was not assessed. Future studies should determine whether CS–BC scaffolds can promote spontaneous differentiation of progenitor cells in non-osteogenic environments. Third, the scaffold’s high hydrophilicity and fluid retention capacity posed challenges in certain analytical procedures, such as dye staining, which may limit specific visualization techniques. Most critically, no in vivo experiments were conducted; thus, the long-term biodegradation behavior, host tissue responses, and overall efficacy in bone regeneration remain uncertain. Given the slow or incomplete degradation of BC reported in previous animal models, further in vivo investigations are crucial to elucidate degradation kinetics and host integration. Understanding these mechanisms in the context of bone defect models will be essential for optimizing scaffold performance and translating these findings toward clinical application in bone tissue engineering.

## 5. Conclusions

This study successfully developed a CS–BC composite scaffold through the SCPL method, with NaOH/urea solution as the solvent and NaCl particle as a porogen. The results of in vitro biodegradable and biocompatible studies indicate that the CS–BC composite scaffold could be degraded, was non-toxic, and supported the cellular activity of MC3T3-E1 cells. The findings of this study suggested that CS–BC composite scaffolds have potential for clinical applications in bone regeneration. In the future, this composite scaffold should be evaluated by in vivo studies to determine its efficacy in bone tissue repair.

## Figures and Tables

**Figure 1 cells-14-00723-f001:**
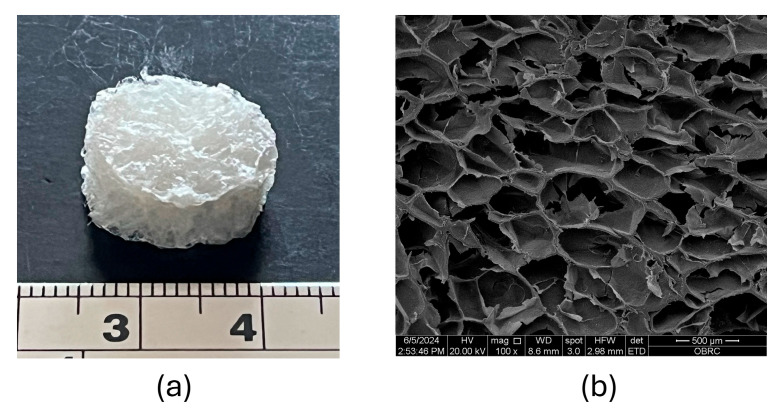
Photograph of the freeze-dried CS–BC composite sponge (**a**) and SEM image of the surface morphology and porosity of the cross-sectional scaffold (**b**) at 100× magnification.

**Figure 2 cells-14-00723-f002:**
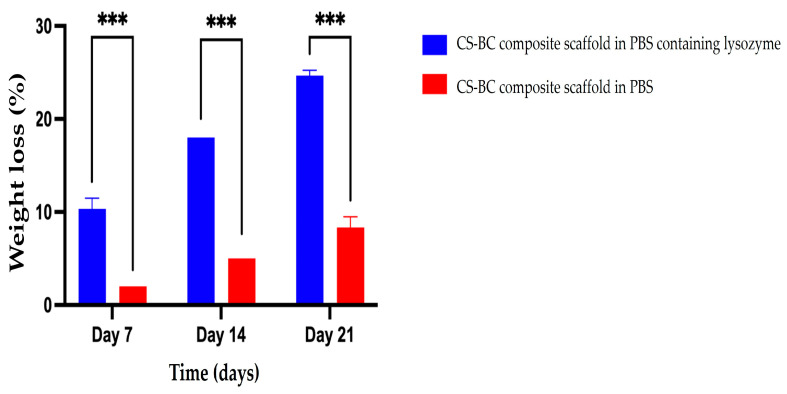
Percentage of weight loss of CS–BC composite scaffolds following immersion in PBS with and without lysozyme at different time intervals. Statistical significance is indicated as *** *p* < 0.001.

**Figure 3 cells-14-00723-f003:**
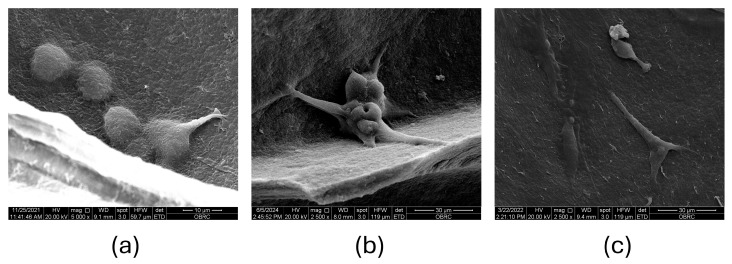
The attachment of MC3T3-E1 cells on the CS–BC composite scaffolds was evaluated through SEM images: day 1 at 5000× magnification (**a**), day 3 at 2500× magnification (**b**), and day 7 at 2500× magnification (**c**).

**Figure 4 cells-14-00723-f004:**
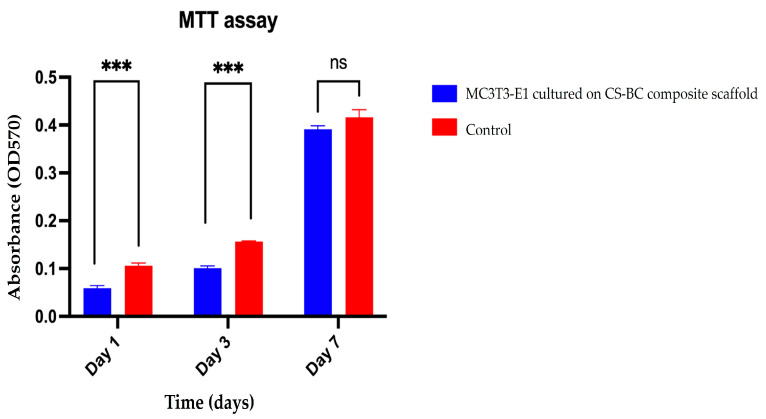
MC3T3-E1 cell metabolic activity on CS–BC composite scaffolds were compared to the control group (scaffold-free) on days 1, 3, and 7 (statistical significance indicated as *** *p* < 0.001, ns: not significant).

**Figure 5 cells-14-00723-f005:**
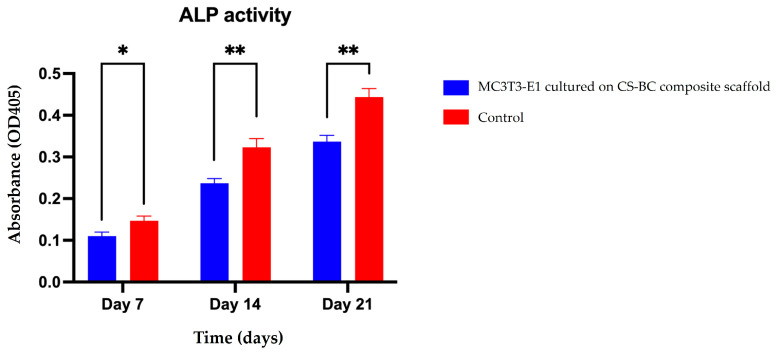
ALP activity of MC3T3-E1 cells cultured on CS–BC composite scaffolds and scaffold-free control groups at days 7, 14, and 21 (statistical significance indicated as * *p* < 0.05, ** *p* < 0.01).

**Figure 6 cells-14-00723-f006:**
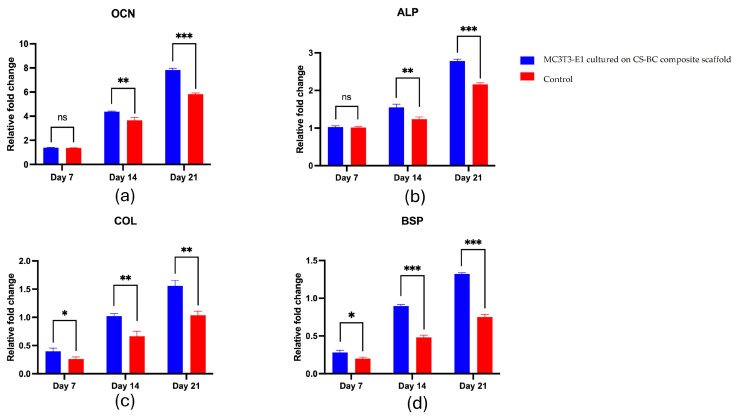
RT-qPCR analysis for *OCN* (**a**), *ALP* (**b**), *COL-1* (**c**), and *BSP* (**d**) expression in MC3T3-E1 cells cultured on CS–BC composite scaffolds and scaffold-free control groups at days 7, 14, and 21 (statistical significance indicated as * *p* < 0.05, ** *p* < 0.01, *** *p* < 0.001, ns: not significant).

**Table 1 cells-14-00723-t001:** Osteogenic-related genes and primer sequences for RT-qPCR analysis in MC3T3-E1 cells.

Gene Names	Forward/Reverse Primer Sequences
*OCN*	5′-TGACCTCACAGATCCCAAGCC-3′/5′-ATACCGTAGATGCGTTTGTAGGC-3′
*ALP*	5′-CCTTGCCTGTATCTGGAATCCT-3′/5′-GTGCAGTCTGTGTCTTGCCTG-3′
*COL-1*	5′-GGGTCTAGACATGTTCAGCTTTGTG-3′/5′-ACCCTTAGGCCATTGTGTATGC-3′
*BSP*	5′-CCTCCTCTGAAACGGTTTCCA-3′/5′-TCTGCATCTCCAGCCTCCTTG-3′
*GAPDH*	5′-AGGTCGGTGTGAACGGATTTG-3′/5′-GGGGTCGTTGATGGCAACA-3′

## Data Availability

The data are contained within the article.

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
