# Peer review of "Investigation of Biodegradation and Biocompatibility of Chitosan–Bacterial Cellulose Composite Scaffold for Bone Tissue Engineering Applications"

_cells, 2025, doi:10.3390/cells14100723_

Round 1

Reviewer 1 Report

Comments and Suggestions for Authors

This manuscript primarily investigates the potential application of CS–BC composite materials in bone tissue engineering, focusing on their biodegradability and biocompatibility. However, the study lacks any significant breakthrough in terms of material innovation and does not convincingly demonstrate novel mechanisms or pronounced performance advantages over existing approaches. The main concerns are outlined below.

Innovation and Significance:

  1. Although CS and BC each hold research value in the biomedical field, their combination, as well as their application in bone tissue engineering, has already been extensively reported in the literature. The manuscript does not clearly show any major improvement or distinctive innovation over existing material systems.
  2. The study briefly mentions the structural similarities between CS and BC and their potential synergistic effects. However, it provides insufficient characterization or mechanistic discussion to highlight any effective synergistic enhancement between the two components.
  3. While the SCPL method is used, it is unclear what aspects distinguish this method from other conventional fabrication processes. For instance, does it achieve significant differences in pore structure, mechanical properties, degradation rate, or biological function that warrant special attention?

Experimental Design:

  1. Although the text repeatedly mentions that the scaffold has “good mechanical strength,” there are no actual data or figures (e.g., compression modulus, bending tests, or other mechanical evaluations) to verify this claim. Merely describing it in words is insufficient for convincing readers.
  2. The degradation and cell experiments often only use a “no lysozyme” or “empty culture plate” More comprehensive comparisons, such as a pure chitosan scaffold or a pure bacterial cellulose scaffold, are required to demonstrate the added benefits of combining CS and BC. For instance, exactly how much performance improvement does bacterial cellulose bring when added to a plain chitosan scaffold?
  3. The data provided rely solely on MC3T3-E1 cells (mouse pre-osteoblasts) cultured on 2D plates or 3D scaffolds for in vitro proliferation and osteogenic gene expression. This alone is insufficient to substantiate the actual performance of the material in vivo. An animal study (e.g., bone defect model) is recommended to demonstrate its efficacy in tissue repair.
  4. Although the conclusion states that “the CS-BC composite scaffold effectively promotes MC3T3-E1 osteogenic differentiation,” Section 3.4 does not offer convincing comparative data. The authors mainly note that ALP levels increase over time but do not address whether this is superior to the blank or other controls. In fact, from Figure 5, the ALP activity appears better in the control group, which also conflicts with the mRNA data in Figure 6. Additional standard staining assays (e.g., ALP staining, Alizarin Red) are needed to visually confirm osteogenic differentiation.
  5. The common approach for measuring ALP activity involves a colorimetric or chromogenic substrate method (e.g., pNPP or BCIP/NBT). However, results typically need to be normalized to cell number or total protein content to accurately compare different conditions or time points. Merely presenting the raw OD values, without clarifying whether cell counts or protein levels were accounted for, makes it difficult to conclude the true trend in ALP activity.

Other Points:

  1. Terms like “ALP” and “MC3T3-E1” appear for the first time in the Abstract without their full names or brief explanations.
Comments on the Quality of English Language

fine.

Author Response

Author’s Response Letter

Dear Reviewer and Editor-in-Chief

Firstly, we thank the Editor and Reviewers who spent time reviewing our manuscript. For this revision round, we responded to the comments from the Reviewers (as shown in the table below). Moreover, any change in the text is clearly shown in the file “revised manuscript.docx”.

Best regards

Somchai Yodsanga

Reviewer 2 Report

Comments and Suggestions for Authors

This paper investigates the biodegradation and biocompatibility of chitosan-bacterial cellulose (CS-BC) composite scaffold for bone tissue engineering. The authors claim the feasibility of CS-BC composite scaffold for bone tissue engineering by illustrating data regarding its biodegradation and biocompatibility. However, I think that it is necessary to add the comparison data with other representative scaffolds. In addition, direct biological data regarding mechanism, that is, why the CS-BC composite scaffold promoted the osteoblastic differentiation, were missing. Furthermore, there are a lot of papers on CS-BC composite scaffold. The novelty of this paper should be clarified for the publication. Taken together, I have to say that this paper is not suitable for the publication in Cells.

Author Response

(The authors gave the same response as above.)

Reviewer 3 Report

Comments and Suggestions for Authors

The manuscript described the biocompatibility, degradation and cell metabolism over Chitosan-Bacterial Cellulose Composite Scaffold. The experiments were well conducted. However some points could be improved to make the manuscript clearer. 

Keywords: by means of indexing It is not advisable to repeat words in Title and keywords.

Methods:

  • line 135 - how long the scaffold was dried
  • MTT assay can not evaluate cell proliferation but cell metabolism, in other words, increase of cell metabolism is not direct related to cell proliferation
  • line 195 - GAPDH is not the more stable endogenous gene, and can mask ΔΔCt analysis. Usually the stability of this gene is check in the experiment conditions, or the average of three endogenous genes are used

Results

  • phrase between lines 229-231 is much more methodology than results
  • apparently control group is used for degradation, differentiation and gene expression assay, but they different controls and not always the same for the three assay. To be clear, "control group" could be changed by the real condition.
  • line 252: microvilli-like projections is not properly used. Microvilli are a special apical membrane structure on epithelial cells, and is used for absorption. Maybe podia/pseudopodia could be used.
  • lines 261-273: MTT assay can not evaluated cell proliferation, cell viability neither cell adhesion.
  • all figures legends could be improved to explain the images

Discussion

  • line 341: Lysozyme is a enzyme produced majorly by epithelial cells in mucosas. This assay does not explain the scaffold degradation on a bone graft, because it will happens majorly by phagocytosis. How correlated lysozyme digestion and graft degradation/absorption?

Conclusion

  • lines 388-392: is much more a results summary not a conclusion
  • line 394: experiments do not evaluate cell division or proliferation, then it could not be concluded.

Author Response

(The authors gave the same response as above.)

Round 2

Reviewer 2 Report

Comments and Suggestions for Authors

I do not think that the authors have addressed the reviewer's comment well.  The authors are required to clearly demonstrate the comparison data with other representative scaffolds, direct data regarding mechanisms undelying feasibility of scaffold for bone tissue engineering, and novelty of the scaffold. If the authors do not possess these data, the authors have to clarify and discussthis point  by quoting appropriate literatures. Therefore, I still have to say that this paper is not suitable for the publication in Cells.

Author Response

Comments: I do not think that the authors have addressed the reviewer's comment well. The authors are required to clearly demonstrate the comparison data with other representative scaffolds, direct data regarding mechanisms undelying feasibility of scaffold for bone tissue engineering, and novelty of the scaffold. If the authors do not possess these data, the authors have to clarify and discuss this point by quoting appropriate literatures. Therefore, I still have to say that this paper is not suitable for the publication in Cells.

Response: We sincerely thank the reviewer for the thoughtful comments and highlighting critical aspects requiring further clarification. We have substantially revised the Discussion section to address your concerns better.

            We have now included a detailed comparison of our CS–BC composite scaffold with representative scaffolds commonly used in bone tissue engineering, focusing on mechanical strength, biocompatibility, biodegradability, and osteoinductive properties. Data from previously published literature support this comparison and demonstrate our scaffold's relative performance and advantages.  We have also elaborated on the possible mechanisms underlying the scaffold’s potential in bone regeneration, including the synergistic effect of chitosan and bacterial cellulose in promoting osteogenic differentiation and cellular attachment.  In addition, we have emphasized the novelty of our scaffold fabrication method, which differs from conventional techniques by utilizing a hybrid process that enhances the homogeneity and structural integrity of the composite.

            Although in vivo data on biodegradability and vascularization are not included in this study, we have acknowledged this limitation and incorporated a thorough discussion of relevant studies supporting the degradable and angiogenic potential of CS- and BC-based scaffolds.  We hope these revisions sufficiently address the reviewer’s concerns and demonstrate our work's scientific value and novelty.

Round 3

Reviewer 2 Report

Comments and Suggestions for Authors

The reviewer agrees the publication of this manuscript in Cells.